# Photorespiration: The Futile Cycle?

**DOI:** 10.3390/plants10050908

**Published:** 2021-05-01

**Authors:** Xiaoxiao Shi, Arnold Bloom

**Affiliations:** Department of Plant Sciences, University of California Davis, Davis, CA 95616, USA; xxsshi@ucdavis.edu

**Keywords:** photorespiration, oxygenation, photosynthesis, metal cofactor, atmospheric CO_2_, climate change, crop yield, metabolic interactions, kinetics

## Abstract

Photorespiration, or C_2_ photosynthesis, is generally considered a futile cycle that potentially decreases photosynthetic carbon fixation by more than 25%. Nonetheless, many essential processes, such as nitrogen assimilation, C_1_ metabolism, and sulfur assimilation, depend on photorespiration. Most studies of photosynthetic and photorespiratory reactions are conducted with magnesium as the sole metal cofactor despite many of the enzymes involved in these reactions readily associating with manganese. Indeed, when manganese is present, the energy efficiency of these reactions may improve. This review summarizes some commonly used methods to quantify photorespiration, outlines the influence of metal cofactors on photorespiratory enzymes, and discusses why photorespiration may not be as wasteful as previously believed.

## 1. Introduction

Photorespiration involves the oxygenation of ribulose-1,5-bisphosphate (RuBP) to form 3-phosphoglycerate (3PGA) and 2-phosphoglycolate (2PG) and the subsequent carbon oxidation pathways that release CO_2_ under light conditions [1,2,3,4,5]. Because it produces 2PG, a compound “toxic” to many enzymes in photosynthetic metabolism, and oxidizes organic carbon without seemingly generating ATP, photorespiration is generally considered a wasteful process. The following sections examines how the photorespiratory pathway converts 2PG into glycolate, the only carbon source for the photosynthetic carbon oxidation cycle [6], a cycle that together with nitrogen assimilation, C_1_ metabolism, and sulfur assimilation generates essential amino acids and intermediate compounds [7]. Moreover, the three enzymes involved in the initial photorespiratory steps within chloroplasts—Rubisco, malic enzyme, and phosphoglycolate phosphatase—have metal binding sites that accommodate either Mg^2+^ or Mn^2+^, and balance between the binding of these enzymes to Mg^2+^ or Mn^2+^ may shift the relative rates and energy efficiencies of photosynthesis and photorespiration [8].

## 2. Photosynthesis vs. Photorespiration

### 2.1. Rubisco

Atmospheric CO_2_ concentration has increased more than 20% during the past 35 years [9]. The major sink for this CO_2_ is the approximately 258 billion tons per year that photosynthetic organisms convert into organic carbon compounds through carbon fixation via the Calvin–Benson pathway [10]. This pathway begins with Rubisco (Ribulose 1,5-bisphosphate carboxylase–oxygenase), the most abundant protein on the planet [11].

Rubisco comes in three forms [12]:

Form I, which is found in cyanobacteria, proteobacteria, chlorophyte algae, heterokont algae, and haptophyte algae, and higher plants, is the most common [13,14]. It is a hexadecamer containing eight identical large subunits (~55,000 *M*_r_), each with a metal-binding site, and eight small subunits (~15,000 *M*_r_). The large subunits are coded by a single plastomic gene, whereas the small subunits are coded by a nuclear multigene family that consists of 2 to 22 members, depending on the species [15]. Complex cellular machinery is required to assemble this form of Rubisco and to maintain its activity [16]. Form I Rubisco, until recently, had resisted all efforts to generate a functional holoenzyme in vitro or upon recombinant expression in *E. coli* [17].

Form II Rubisco, found in proteobacteria, archaea, and dinoflagellate algae, contains one or more isodimers with subunits that share about 30% identity to the large subunit of Form I Rubisco [8].

Form III Rubisco, found in archaea, has one or five isodimers composed of subunits homologous to the large subunit of Form I Rubisco [8].

Form II and Form III Rubisco show greater similarity in their primary sequence to one another than either do to the large subunit of Form I Rubisco [8].

All three forms of Rubisco catalyze not only the reaction in which the carboxylation of the five-carbon sugar RuBP generates two molecules of the three-carbon organic acid 3-phosphoglycerate (3PGA), but also an alternative reaction in which oxidation of RuBP generates one molecule of 3PGA and one of 2PG (Figure 1) [8]. The carboxylation pathway of photosynthesis expends 3 ATP and 2 NADPH per RuBP regenerated and produces a carbon in hexose [18], whereas the oxygenation pathway of photorespiration reportedly expends 3.5 ATP and 2 NADPH per RuBP regenerated but produces no additional organic carbon [19,20].

Rubisco must be activated before it can carboxylate or oxygenate RuBP. Activation of the three forms of Rubisco involves binding of Mn^2+^ or Mg^2+^ [21,22]. Binding of Mg^2+^ requires carbamylation of Rubisco by the addition of CO_2_. One histidine at the active site of Rubisco rotates into an alternate conformation, resulting in a transient binding site where Mg^2+^ is partially neutralized by the conversion of two water molecules to hydroxide ions and coordinated indirectly by three histidine residues through the water molecules. Subsequently, the hydroxide ions cause a lysine residue at the active site to become deprotonated and rotate 120 degrees into the *trans* conformer, which brings its nitrogen into close proximity to the carbon of CO_2_, allowing for the formation of a covalent bond that produces a carbamyl group. This carbamyl group causes the Mg^2+^ ion to transfer to a second binding site, after which the histidine that first rotated returns to its original conformation [23]. It is unclear whether binding Mn^2+^ follows a similar mechanism and whether it requires an activator CO_2_ to be bound first [21,22]; hence, understanding the mechanism of Mn^2+^ binding to Rubisco is important to future research on Rubisco kinetics. During in vitro studies, Rubisco is often activated at pH 8.0 in the presence of CO_2_ and either Mg^2+^ or Mn^2+^.

Rubisco can also bind to other metals. When bound to Fe^2+^, Ni^2+^, Cu^2+^, Ca^2+^, or Co^2+^, Rubisco may exhibit some carboxylase and oxygenase activity [24]. For example, one study found that Rubisco from *R. rubrum*, when bound to Co^2+^, was incapable of carboxylation but still capable of oxygenation [24]. Another study found that Rubisco from spinach performed both carboxylation and oxygenation when bound to Ni^2+^ or Co^2+^ [25]. When bound to some other metal ions, including Cd^2+^, Cr^2+^, and Ga^2+^, Rubisco cannot catalyze either carboxylation or oxygenation [24]. Although it is known that the metal ion plays a role in stabilizing the activator carbamate and determining the active site’s structure, its effect upon the reactions catalyzed by Rubisco is still not completely understood. One hypothesis is that Mg^2+^, because of its electron-withdrawing properties, polarizes the C2 carbonyl of RuBP, which favors the removal of the C3 proton and thereby contributes to enolization [21].

NADPH complexes strongly with Rubisco and acts as an effector molecule to maintain the Rubisco catalytic pocket in an open confirmation that more rapidly facilitates CO_2_-Mg^2+^ activation when CO_2_ and Mg^2+^ are present in suboptimal concentrations [26,27,28,29]. The crystal structure of Rubisco with both Mg^2+^ and NADPH as ligands indicates that NADPH binds to the catalytic site of Rubisco through metal-coordinated water molecules [26]. The activated enzyme catalyzes either carboxylation or oxygenation of the enediol form of the five-carbon sugar ribulose-1,5-bisphosphate (RuBP) [14,21,22,30,31].

### 2.2. Balance between Carboxylation and Oxygenation and Metal Cofactors

Several factors alter the balance between Rubisco carboxylation and oxygenation and, thereby, alter the relative rates of photosynthesis and photorespiration. These include the concentrations of CO_2_ and O_2_ at the active site of Rubisco, the specificity of the enzyme for each gas, and whether the enzyme is associated with Mg^2+^ or Mn^2+^ [32]. These divalent cations share the same binding site in Rubisco [14,22,33], and in tobacco, Rubisco associates with both metals and rapidly exchanges one metal for the other [32]. Nonetheless, nearly all recent studies on the photosynthetic and photorespiratory pathways have been conducted in the presence of Mg^2+^ and absence of Mn^2+^ [8]. Rubisco binding of Mg^2+^ accelerates carboxylation, whereas binding of Mn^2+^ slows carboxylation [25,34,35,36,37,38]. Chloroplastic Mg^2+^ and Mn^2+^ activities seem to be regulated at the cellular level because in isolated tobacco chloroplasts, activities of the metals were directly proportional to their concentrations in the medium [32]. The thermodynamics of binding Mg^2+^ to Rubisco were similar for enzymes isolated from a Form I and a Form II species [32]. By contrast, the thermodynamics of binding differed greatly between the two Rubisco forms when the enzymes were associated with Mn^2+^ [32].

Mg^2+^ and Mn^2+^ have nearly identical ionic radii but highly disparate electron configurations: Mg^2+^ (1s^2^2s^2^2p^6^ or [Ne]) has a very stable outer shell [8], whereas Mn^2+^ has five unpaired d electrons (1s^2^2s^2^2p^6^3s^2^3p^6^3d^5^ or [Ar]3d^5^) that are susceptible to many redox reactions. An aerated solution of activated Mn^2+^-Rubisco exhibits a long-lived chemiluminescence when RuBP is added [39,40]. This chemiluminescence was attributed to a spin-flip within the Mn^2+^ 3d manifold, leading to an excited quartet (*S* = ^3^/_2_) d^5^ electronic configuration that decays over the course of 1 to 5 min back to the sextet (*S* = ^5^/_2_) ground state electronic configuration [39]. Excited states are intrinsically better oxidants and reductants (larger reduction/oxidation potentials) than their corresponding ground states [41,42,43]; thus, the observed chemiluminescence opens the possibility that the RuBP-O_2_-Mn^2+^—Rubisco excited state may be quenched via electron transfer. Consequently, the liberated reducing equivalent could participate in the reduction of NADP^+^ to NADPH (Figure 2, blue pathway). In this way, oxidation of RuBP via O_2_ may proceed in a spin-allowed manner, while the Mn^2+^ remains “innocent” in the generation of the oxygenated RuBP precursor. Mn^2+^-centered redox may still proceed, with oxidation of excited Mn^2+^ to Mn^3+^ occurring in a manner independent of, but parallel to, substrate oxidation.

In wheat leaves, the ratio of Mn^2+^ to Mg^2+^ contents increased as the CO_2_ levels increased and when the nitrogen source was nitrate rather than ammonium [32]. Nitrate assimilation into amino acids in shoots is heavily dependent on photorespiration, whereas ammonium assimilation is much less so. This indicates that plants shifted to Rubisco Mn^2+^ binding in order to compensate for the slower photorespiration rates and slower amino acid production that would otherwise occur under elevated CO_2_ and nitrate nutrition.

### 2.3. The Photorespiratory Pathway

The 3-phosphoglycerate produced during photorespiration, like that produced during photosynthesis, is converted to triose phosphate and used to regenerate RuBP. On the other hand, 2-phosphoglycolate is converted to glycolate by phosphoglycolate phosphatase. In the peroxisome and mitochondrion, a series of reactions converts glycolate to glycerate, which is ultimately returned to the chloroplast to regenerate RuBP (Figure 2) [8]. In addition to Rubisco, several other chloroplast enzymes in the photorespiratory pathway, including malic enzyme and phosphoglycolate phosphatase, bind either Mg^2+^ or Mn^2+^ [8]. The plastid isoform of malic enzyme in *Arabidopsis* and tobacco catalyzes the reverse pyruvate synthesis reaction (pyruvate + CO_2_ + NADPH → malate + NADP) [44,45]. Phosphoglycolate phosphatase, which is responsible for the hydrolysis of 2-phosphoglycolate to glycolate, binds to and is activated by either metal [46]. Hypothesized is an alternative photorespiratory pathway that increases photorespiration energy efficiency by generating malate (RuBP + O_2_ + CO_2_ + H_2_O → glycolate + malate + 2Pi) when Mn^2+^ binds to these enzymes (Figure 2) [8].

## 3. Estimating Rates of Photorespiration

Many different methods have been employed for estimating rates of photorespiration. The following sections outline the general approach of each method and highlights the assumptions and potential errors in each. The hope is that certain methods might be better suited for assessing the influence of Mn^2+^ vs. Mg^2+^ on the relative rates of oxygenation and carboxylation in situ.

### 3.1. Traditional Methods for Estimating Photorespiration

#### 3.1.1. Post Illumination CO_2_ Burst

This method measures the evolution of CO_2_ from a leaf for 1 to 4 min after turning off the light because glycine metabolism continues longer in the dark than CO_2_ assimilation [47]. The rate of CO_2_ generation is measured by a transient CO_2_ analyzer [48] when the light has just been turned off or at the maximum rate of CO_2_ evolution observed. CO_2_ assimilation, however, does not stop immediately after the light is off. Separating CO_2_ assimilation from the CO_2_ burst effects during this time is difficult, and hence this method underestimates photorespiratory rates [49,50]. This method also fails to consider variations in mitochondrial respiration, leading to overestimates of photorespiratory rates [51].

#### 3.1.2. O_2_ Inhibition of Net CO_2_ Assimilation

This method aims to assess the photorespiration rate from the increase in the CO_2_ assimilation rate after switching from normal to low O_2_ concentrations. Yet, changes in CO_2_ assimilation with O_2_ concentration may derive from components of the photosynthetic pathway other than photorespiration [4]. For example, when starch and sucrose synthesis limit photosynthesis, increasing or decreasing the photorespiration does not affect net CO_2_ assimilation [52].

#### 3.1.3. Photorespiration CO_2_ Efflux into CO_2_-Free Air

This method estimates photorespiration from the CO_2_ efflux rate in CO_2_-free air. A high-O_2_ and low-CO_2_ environment, however, promotes photorespiration [4]. Additionally, a CO_2_-free atmosphere inhibits both the activity of Rubisco [53] and the regeneration of its substrate RuBP [54], leading to underestimates of photorespiration.

#### 3.1.4. Ratio of ^14^CO_2_ to ^12^CO_2_ Uptake

In this method, ^14^CO_2_ and ^12^CO_2_ fluxes are measured after feeding a leaf with ^14^CO_2_ for a short period of time. Gross photosynthesis is estimated from ^14^CO_2_ uptake measured using an ionization chamber attached to an electrometer, while net photosynthesis is estimated from ^12^CO_2_ measured using an infrared gas analyzer. Photorespiration is estimated as the difference between gross and net photosynthesis [55] (Figure 3).

There are several uncertainties associated with this method. The recycling effect on the specific activity of CO_2_ inside the leaf can cause about a 20% error. One must consider the specific activity of CO_2_ inside the leaf to obtain an accurate estimate of the gross photosynthesis rate because CO_2_ efflux through photorespiration dilutes the ^14^C label in the intercellular spaces, decreasing the specific activity of CO_2_. The activity might be even lower at the actual carboxylation site than in the interleaf spaces because of photorespiratory CO_2_ loss [56]. Moreover, Rubisco carboxylation discriminates about 2.9% against ^13^C [57,58] and about 5.5% against ^14^C [4], resulting in errors in estimations of photorespiration rates that exceed 25% [4,57].

### 3.2. Recent Methods for Estimating Photorespiration

#### 3.2.1. Calculation from Kinetics Models

Rubisco reaction kinetics can provide an estimate of the photorespiration rate [4,59]. This method can provide accurate estimates of photorespiration rates if the CO_2_ compensation point in the absence of mitochondrial respiration (**Γ***) being known for a given plant species. The rate of oxygenation, which is assumed to be twice the rate of photorespiration, is given by:$$v_{o}=\frac{A+R_{d}}{\frac{1}{\Phi }-0.5}$$
where *A* is the rate of photosynthetic CO_2_ assimilation, *R_d_* is the rate of respiration other than photorespiration, and:$$\Phi =\frac{v_{o}}{v_{c}}=\frac{2\Gamma^{*}}{C}$$
where *C* is the CO_2_ concentration.

The principal drawbacks of this method are that it does not directly measure photorespiration and depends on estimates of *C* and Γ*. There are several techniques for estimating *C* at the site of Rubisco activity, but estimating Γ* is more difficult. Values for Γ* are known for only a few species, and depend on estimates of kinetic parameters, which themselves rely on estimates of photorespiration [59].

#### 3.2.2. CO_2_ Efflux into ^13^CO_2_-Air

The gas exchange method is based on the FvCB (Farquhar, van Caemmerer, and Berry) model [4,59]. First, ambient air is rapidly replaced with air containing ^13^CO_2_ and no ^12^CO_2_. The levels of released ^12^CO_2_ can be measured either using an infrared gas analyzer or a membrane inlet mass spectrometer. Because the rate of ^12^CO_2_ release includes both photorespiration and mitochondrial respiration, additional effort is needed to separate these effects. For example, in one approach, the rate of ^12^CO_2_ release is calculated as:$$R_{12_{C}}=\frac{F\left({}^{12}C_{R}-{}^{12}C_{S}\frac{1-W_{R}}{1-W_{S}}\right)}{a}$$
where *F* is the gas flow rate, ^12^*C_R_* and ^12^*C_S_* are the mole fractions of ^12^CO_2_ in the chamber without and with a leaf, *W_R_* and *W_S_* are the corresponding water mole fractions, and *a* is the illuminated leaf area in the chamber [60]. To divide this quantity into photorespiration and mitochondrial respiration, the air is replaced with air containing 10,000 ppm ^13^CO_2_ and the concentration of ^13^CO_2_ over 2 min is fitted to an exponential curve. The mitochondrial respiration *R_d_* is taken to be the rate of ^12^CO_2_ release after 2 min.

This method can provide estimates of both carboxylation and oxygenation if one assumes that the rate of mitochondrial respiration (*R*_d_) is not affected by the sudden high CO_2_ concentration, that 0.5 CO_2_ is generated per oxygenation reaction when the CO_2_ released per oxygenation varies widely with temperature and light level and among species [7], and that leaves do not naturally contain any ^13^C [59]. If intracellular reassimilation is significant and it often is [60], substantial errors in the estimate can result. These errors can be accounted for by monitoring the release of ^12^CO_2_ after switching from ambient air to air with a high concentration of ^13^CO_2_; however, high CO_2_ concentrations could affect mitochondrial respiration and thus produce error in the estimate of photorespiration. The presence of naturally occurring ^13^C also generates additional errors [59].

#### 3.2.3. Labelling of Photosynthates with ^14^C

Leaves at a photosynthetic steady state are exposed to ^14^CO_2_ for different lengths to label primary and stored photosynthates. Exposing the leaf to an ambient concentration of ^14^CO_2_ for 10 to 15 min will label primary photosynthates, such as the metabolites from the Calvin cycle, glycolate cycle, and intermediates of starch and sucrose synthesis and of glycolysis [44]. Longer exposures (2 to 3 h) will label stored photosynthates, such as starch, sucrose, fructans, and vacuolar acids. ^14^CO_2_ efflux into different backgrounds containing various combinations of O_2_ and CO_2_ concentrations provides an estimate of photorespiration [61,62]. Four different backgrounds are used: first, 21% O_2_ and ambient CO_2_ to measure the steady-state release of CO_2_ from both photosynthesis and photorespiration; second, 1.5% O_2_ and ambient CO_2_ to measure the rate of photorespiration only; third, 21% O_2_ and 30,000 μmol/mol CO_2_ to limit CO_2_ reassimilation; and fourth, 21% O_2_ with no CO_2_ to measure the specific radioactivity of CO_2_ efflux [63].

The assumptions for this method are that all photosynthates must be labeled during the labeling time frames and that *R*_d_ is not affected by the percentage of O_2_ in the air. A recent report indicated that *R*_d_ was actually lower at a lower O_2_ concentration (2%) than at an ambient concentration (21%) [64]. One also has to assume that the mitochondrial respiration (R_d_) value does not change upon transient exposure to high CO_2_ levels.

#### 3.2.4. Measuring Photorespiratory Ammonia

Photorespiration generates NH_3_ in addition to CO_2_ during the conversion from glycine to serine in mitochondria [65]. Adding glutamine synthetase (GS) inhibitors methionine sulphoximine [35] or phosphinothricin [66] prevents ammonia reassimilation in chloroplasts, and NH_3_ subsequently accumulates in the leaf. The advantages of this method also include the prevention of CO_2_ refixation and uncertainties in *R*_d_ values under the experimental conditions [35,66]. This approach, however, depends on several assumptions: (1) The GS inhibitors do not inhibit photorespiration, and (2) they can prevent NH_3_ refixation completely.

Other factors might limit the diffusion of NH_3_ out of the leaves, leading to an underestimation of photorespiration [59]. GS inhibitors will disrupt the C_2_ cycle under photorespiratory conditions, and glycolate will rapidly accumulate, which in turn will inhibit photosynthesis. Feeding the plant an amino acid donor, such as glutamine, together with GS inhibitors will help minimize this inhibition effect [66,67].

Quantification of ammonia poses some challenges. The commonly used ion chromatography method to quantify NH_4_^+^ may overestimate the amount of NH_4_^+^ because methylamine, ethylamine, ethanolamine, and some non-protein amino acids co-elute with NH_4_^+^. Degradation of labile nitrogen metabolites in leaf extract, xylem sap, and apoplastic fluid to NH_4_^+^ during extraction will cause further overestimation of NH_4_^+^ levels [68].

#### 3.2.5. Measuring ^18^O_2_ Consumption and Labeled Metabolites

Replacing ambient air in a chamber containing a leaf with air containing ^18^O_2_ provides another estimate of the photorespiration rate. A mass spectrometer measures levels of ^16^O_2_ and ^18^O_2_. The rate of oxygenation is estimated as:$$v_{o}=\frac{2}{3}\left({}^{18}O_{2}\;uptake\;in\;light-{\;}^{18}O_{2}\;uptake\;in\;dark\right)$$
and carboxylation as:$$v_{c}={\;}^{16}O_{2}\;evolution-v_{o}$$
Ref. [69,70].

Unfortunately, this method cannot separate photorespiration from other light-dependent O_2_-consuming processes, such as light-dependent differences in the rate of mitochondrial respiration [46,48]. To diminish these errors, the mass spectrometer can quantify ^18^O-labeled metabolites, such as glycolate, glycine, and serine; with several assumptions about the photorespiratory pathway, such as the pool sizes of the labeled metabolites [49], one can then use the amounts of labeled metabolites to calculate the photorespiration rate [71,72].

#### 3.2.6. NMR Measurements on ^13^C-Labeled Metabolites

This method requires that plants receive fertilizer labeled with ^15^N and that leaves subsequently be exposed to ^13^CO_2_. Rotational-echo double resonance (REDOR) detects ^13^C within two covalent bonds of ^15^N and thus assesses the formation of organic nitrogen metabolites labeled with ^13^C [59,73]. The ratio of ^13^C-labeled to unlabeled phosphorylated Calvin–Benson cycle metabolites between 2 and 4 min after exposure to ^13^CO_2_ indicates the ratio of photosynthesis to photorespiration [50]. This assumes that metabolites produced from photosynthesis are fully labeled in less than 2 min after being exposed to ^13^CO_2_ and that those produced from photorespiration do not become labeled until after 4 min. These assumptions may lead to errors because photosynthesis may re-assimilate some of the ^12^CO_2_ generated by photorespiration and because photorespiration may produce intermediates labeled with ^13^C in less than 2 min [60]. Furthermore, this method is based on the premise that photorespiration releases one CO_2_ for every two oxygenations, when the CO_2_ released per oxygenation varies widely with temperature and light level and among species [7].

#### 3.2.7. Quantification of 2-Phosphoglycolate (2PG) and Photorespiratory Metabolites by Mass Spectrometry

This method uses LC-MS/MS to measure directly the first intermediate, 2PG, of photorespiration when Rubisco oxygenates RuBP, and GC-MS to measure other photorespiratory metabolites. In the LC-MS/MS portion, 2PG is separated from other molecules in three steps: First, liquid chromatography separates 2PG based on its physiochemical properties; second, mass spectrometry separates 2PG based on its *m/z* ratio; and third, mass spectrometry separates 2PG based on its *m/z* ratio after being fragmented [74]. Readings from the LC-MS/MS samples are compared with 2PG standard solutions [75]. Additionally, GC-MS is used to quantify additional photorespiratory metabolites, such as glycolate, glyoxylate, glycine, serine, hydroxypyruvate, and glycerate [74,76].

This approach has estimated photorespiratory rates in plant mutants deficient in expression of genes coding for photorespiratory enzymes. The gaseous environment of the aerial part of the plant, but not the root, was altered before experimentally determining the changes in the metabolite (2PG) content [77,78,79,80,81].

This method has several problems [74,82,83,84,85]. First, non-volatile salts and metabolites were deposited at the inlet of MS/MS after eluting from the LC step, which is very common when using anion-exchange chromatography [84]. Second, numerous metabolites eluted from the LC step had overlapping and asymmetrical peaks resulting from the matrix effect (interference in the ionization between compounds with similar elution times) [82,85], which significantly affects the sensitivity and accuracy of the measurements on a specific metabolite, such as 2PG. Third, post-harvest changes in metabolite concentrations can severely affect the quantification of 2PG [74,83]. Fourth, the GC-MS step is not targeted and therefore is potentially prone to error if other compounds with a similar molar mass as the photorespiratory metabolites are present [76].

#### 3.2.8. CO_2_ Labeling and MS Analysis

Isotopically nonstationary metabolic flux analysis (INST-MFA) can trace ^13^C-labeled photorespiratory metabolites in plants exposed to ^13^CO_2_ to assess the photorespiration rate [86,87,88,89,90]. Monitoring the isotope incorporation in downstream metabolites over time assesses the relative contributions of different pathways after administration of the tracer. The turnover rates of each enzyme determine the labeling dynamics (Figure 4). Mathematical metabolic models specific for each pathway are often used to enumerate mass and isotopomer balances and ensure atoms’ conservation within the system. The models’ proposed metabolic fluxes are compared with those measured experimentally, and differences are minimized with each subsequent iteration.

The INST-MFA approach presents several challenges. A minimum of three sample time points is needed for precise measurements of metabolic fluxes [91]. This makes experimental design more complex and time-consuming. To ensure accurate and precise measurements, the pool size for each component of a metabolic pathway has to be very specific. Absolute quantification of intracellular pool sizes, however, is not yet possible even with pool size measurements made with optimized mathematical modeling [91]. A second challenge of this approach is isotopic transients. Some intracellular metabolites can exhibit short isotopic transients that last only for a few minutes or seconds. Rapid sampling and quenching have to be achieved to obtain precise and meaningful INST-MFA measurements [92].

#### 3.2.9. Micro-Optode Measurement of O_2_ Consumption

We have been conducting direct oxygenation rate measurements using a needle-type O_2_ micro-optode to examine the effects of metal cofactors on Rubisco photorespiration reactions. In this instrument, a polymer optical fiber transmits the excitation wavelength to the tip of the sensor and at the same time transmits the fluorescence response of an oxygen-sensitive dye that is immobilized in a polymer matrix at the tip. The rate of oxygenation can be calculated easily by comparing the amount of quenching of the excitation light by dissolved O_2_. The micro-optode has a 50–70-µm tip diameter, which makes it possible for a micro-scale setup, such as in a micro-cuvette or plate. The most important advantages for this type of sensor are that the micro-optode does not consume O_2_ in contrast to the other commonly used O_2_ sensors, such as a Clark electrode [93,94]; it has no stirring sensitivity; and it is resistant to most corrosive environments. The micro-optode also works in both gas (%O_2_) and liquid phases (DO), which makes it possible to measure O_2_ exchanges accurately up to 250% air O_2_ saturation in intact plant leaves, bioreactors, cell cultivation, microtiter plates, and many general oxygen measurements in liquids [95,96,97,98,99,100].

## 4. Photorespiration and Other Metabolic Pathways

### 4.1. NO_3_^−^ Assimilation

Multiple lines of evidence link shoot NO_3_^−^ assimilation to photorespiration:(a)Elevated CO_2_ or low O_2_ levels inhibited shoot NO_3_^−^ reduction [101].(b)In independent ^14^N and ^15^N labeling experiments, assimilation of either ^14^N–NO_3_^−^ or ^15^N–NO_3_^−^ decreased under CO_2_ enrichment [102].(c)Under elevated CO_2_ conditions, NO_3_^–^ nutrient absorption and organic N accumulation levels in various plant species declined when plants received NO_3_^−^ as a sole N source [102,103,104,105,106].(d)C_3_ plants receiving NO_3_^−^ as their sole N source experienced slower growth under CO_2_ enrichment than those receiving NH_4_^+^ [9,107,108].

In wheat and *Arabidopsis* plants grown under CO_2_ enrichment and receiving NO_3_^−^ containing ^15^N at natural abundance levels*,* shoot tissues became less enriched with ^15^N organic compounds [102,109]: elevated CO_2_ inhibited shoot NO_3_^−^ reduction so it was less limited by nitrate availability, and NO_3_^−^ reductase discriminated more strongly against ^15^N–NO_3_^−^ [110].

The assimilatory quotient (*AQ*) is the ratio of net CO_2_ consumption to net O_2_ evolution in plant shoots [111]. During shoot NO_3_^−^ assimilation, ferredoxin generated from the photosynthetic electron chain reduces NO_2_^−^ to NH_4_^+^ rather than producing NADPH, and so net O_2_ evolution increases without a change in net CO_2_ consumption. Therefore, the change in assimilatory quotient (∆*AQ*) when a plant receives NH_4_^+^ instead of NO_3_^−^ as a sole N source provides an estimate of shoot NO_3_^−^ assimilation [106]. ∆*AQ* decreased as the shoot internal CO_2_ concentration increased in C_3_ plants (Figure 5) [9,104,112,113].

Shoot CO_2_ and O_2_ fluxes at ambient and elevated CO_2_ were contrasted between stages of plant development or genotypes that have significantly different NO_3_^−^ reductase activities in situ (i.e., 36- vs. 48-day-old wild-type *Arabidopsis*, *Arabidopsis* NO_3_^−^ reductase knockout mutants vs. transgenic *Arabidopsis* overexpressing NO_3_^−^ reductase, and NO_3_^–^ reductase-deficient barley mutants vs. wild-type barley) [104,112]. *∆AQ*, a measure of shoot NO_3_^–^ assimilation, differed between these stages of development and genotypes under ambient CO_2_ but not under elevated CO_2_. This indicates that none of the stages of development or genotypes were assimilating NO_3_^−^ under elevated CO_2_ [104,112].

Maximum NO_3_^−^ reductase activity in vitro generally declined under CO_2_ enrichment [105,114]. Nonetheless, shoot NO_3_^−^ reductase activity seldom limits NO_3_^−^ assimilation in planta [115,116]. Accordingly, NO_3_^−^ assimilation significantly declined only in genotypes with mutations that nearly eliminated enzyme activities [104,117,118], and genotypes with 50% higher NO_3_^−^ reductase activities did not assimilate more NO_3_^−^ [119]. Studies that have confused rates of enzyme activities with those of NO_3_^−^ assimilation as a whole have drawn false conclusions [120,121].

One physiological mechanism that may be responsible for the interdependency of photorespiration and shoot NO_3_^–^ assimilation involves the reduction of the Mn^2+^-RuBP complex during oxidation of RuBP. This increases the redox potential of the chloroplast [101], thereby stimulating the production of malate [122,123] and promoting its export from chloroplasts to the cytoplasm. Malate dehydrogenase in the cytoplasm converts malate to oxaloacetate, generating NADH [124,125,126] to empower the initial step of NO_3_^−^ assimilation [127]. Consequently, mutations that alter malate transport or metabolism influence both photorespiration and NO_3_^−^ assimilation [122,128,129].

### 4.2. C_1_ Metabolism

The photorespiratory pathway within mitochondria involve reactions with glycine. In one reaction, serine hydroxymethyltransferase 1 (SHMT1) converts glycine to serine and converts CH_2_-THF (5,10-methylene-tetrahydrofolate) to THF (Figure 2). In the other reaction, the glycine decarboxylase complex reduces NAD^+^ to NADH and catabolizes glycine to CO_2_, NH_3_, and CH_2_-THF (Figure 2). These C_1_ units, in the form of CH_2_-THF, serve as precursors in the synthesis of tetrahydrofuran (THF) derivatives [130,131,132,133]. One derivative of CH_2_-THF, 5-CH_3_-THF, is used to produce methionine, an essential amino acid. Methionine is a powerful antioxidant and is involved in protein synthesis and methylation of DNA, RNA, proteins, phospholipids, and other substrates [132]. In addition, about 5% of the total assimilated carbon in many secondary metabolites, such as glycine betaine, nicotine, and lignin, derive from C_1_ metabolism [131].

### 4.3. Sulfur Assimilation

Photorespiration stimulates sulfur assimilation, although the effects are relatively small. By tracing ^33^S in reactions involved in sulfur assimilation (such as sulfate reduction and synthesis of cysteine), and ^13^C in glycine and serine, a positive linear relationship was derived between relative photorespiration and sulfur assimilation. Sulfur assimilation decreases as photorespiration declines and photosynthesis increases [134].

Cysteine, the major product from sulfur assimilation, uses the sulfur element converted from serine generated from photorespiratory pathways [134,135]. H_2_S, produced from sulfite reduced by sulfite reductase, is incorporated into O-acetylserine (OAS) via a protein complex consisting of serine acetyl transferase and OAS thiol-lyase to form cysteine [135,136]. Cysteine is essential in methionine synthesis, glutathione metabolism, sulfur-rich protein synthesis, glucosinolate biosynthesis, and the synthesis of phytoalexins (Figure 6) [137]. Cysteine is the precursor of methionine through *o*-phosphohomo-serine and homocysteine. Using methyl tetrahydrofolate as a cofactor, homocysteine is methylated by methionine synthase to yield methionine. Cysteine and methionine are the major sulfur contributors found in downstream metabolites, the most important of which is S-adenosyl methionine (SAM), which is a donor in methyl group transfers, transsulfuration, and aminopropylation [135,138].

## 5. Conclusions

Is photorespiration simply a futile cycle? The answer is “no”. Multiple lines of evidence show its crucial role in many plant processes. Despite heroic efforts to suppress photorespiration, disrupting any photorespiratory reaction usually proves detrimental to plants [139,140]. The reassimilation of CO_2_ from photorespiration [60] and the important role played by photorespiration in the acclimation of plants to conditions, such as salinity [141] and elevated CO_2_ [142], are topics that are beyond the scope of this review but nevertheless provide important evidence showing that photorespiration is not a wasteful process. There are many promising directions for further studies on photorespiration; for example, examining Mn^2+^ interactions with Rubisco, further exploring the reassimilation of photorespired CO_2_, and exploring how the biochemical processes related to photorespiration contribute to its role in adaptation to various conditions will probably reveal that plant carbon fixation and respiration is more energy efficient than what has been previously assumed.

## Figures and Tables

**Figure 1 plants-10-00908-f001:**
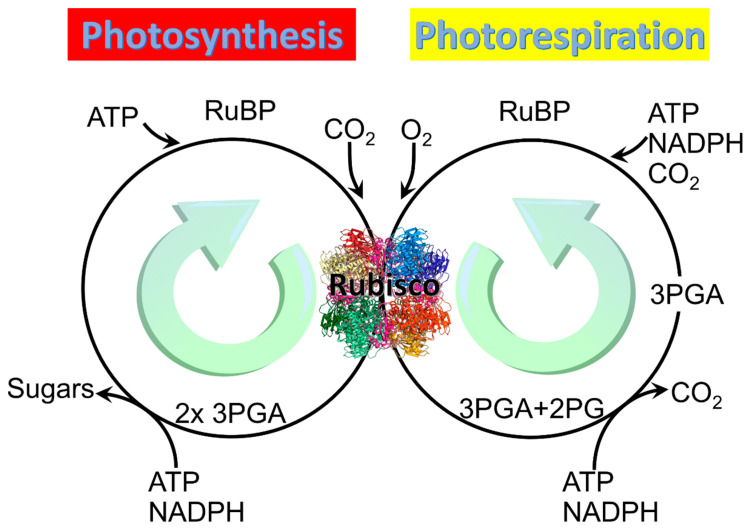
Two main reactions of Rubisco: Photosynthesis and photorespiration. Rubisco structure picture credit: Laguna design / science photo library.

**Figure 2 plants-10-00908-f002:**
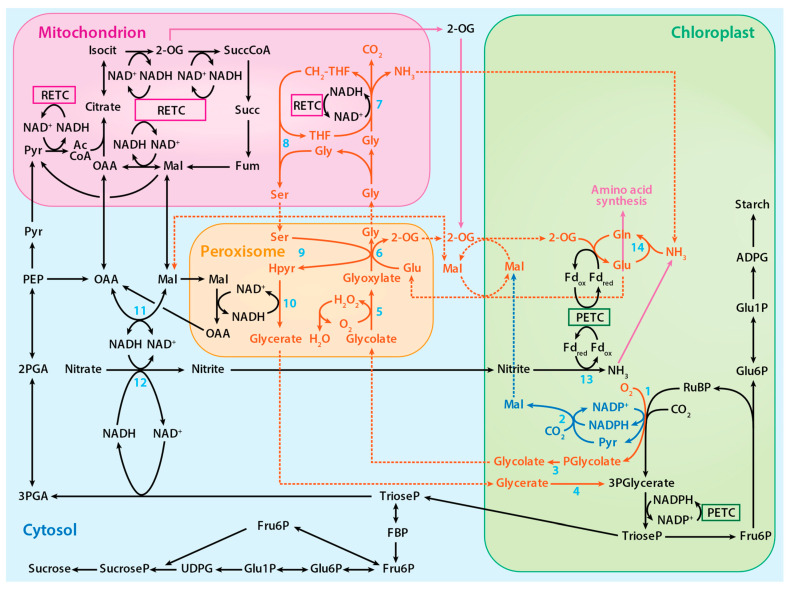
The proposed photorespiratory pathway within the context of photosynthetic carbon and nitrogen metabolism. The solid red lines represent reactions of the photorespiratory pathway, the solid blue lines represent reactions of the proposed alternative photorespiratory pathway, the solid purple lines represent reactions of amino acid synthesis, and the dotted lines represent associated transport processes. Numbered reactions are catalyzed by the following enzymes: 1. Rubisco, 2. Malic enzyme, 3. Phosphoglycolate phosphatase, 4. Glycerate kinase, 5. Glycolate oxidase, 6. Glutamate:glyoxylate aminotransferase, 7. Glycine decarboxylase complex, 8. Serine hydroxymethyltransferase-1, 9. Serine:glyoxylate aminotransferase, 10. Hydroxypyruvate reductase-1, 11. Malate dehydrogenase, 12, Nitrate reductase, 13 Nitrite reductase, and 14. Glutamine synthetase. PETC designates photosynthetic electron transport chain and RETC, respiratory electron transport chain. Adapted from ref. [8]. Copyright 2018 Springer Nature Ltd.

**Figure 3 plants-10-00908-f003:**
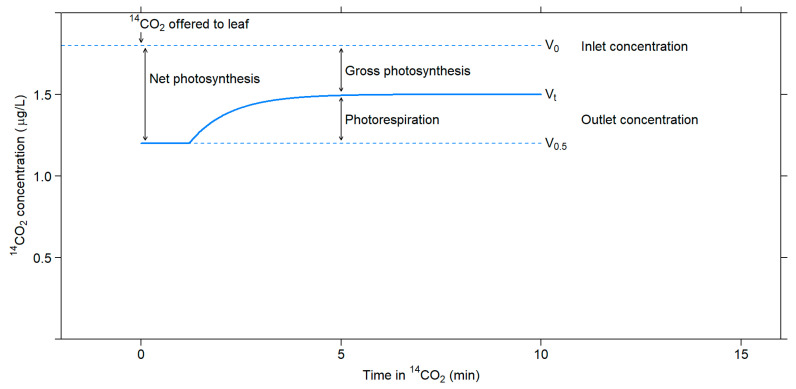
Changes in ^14^CO_2_ concentration that occur upon exposing a leaf in the light to ^14^CO_2_. Adapted with permission from ref. [55]. Copyright 1971 Canadian Science Publishing.

**Figure 4 plants-10-00908-f004:**
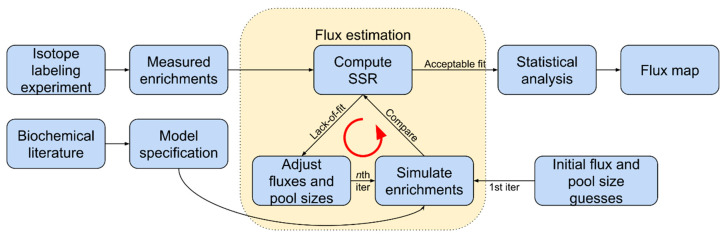
Simplified INST-MFA workflow to estimate flux. Software, such as INCA and OpenMebius, is used in several steps. Fluxes and pool sizes are initially guessed and then adjusted in each iteration, converging upon a flux map that fits measurements of metabolism during the ILE. Adapted with permission from ref. [92]. Copyright 2018 Elsevier Ltd.

**Figure 5 plants-10-00908-f005:**
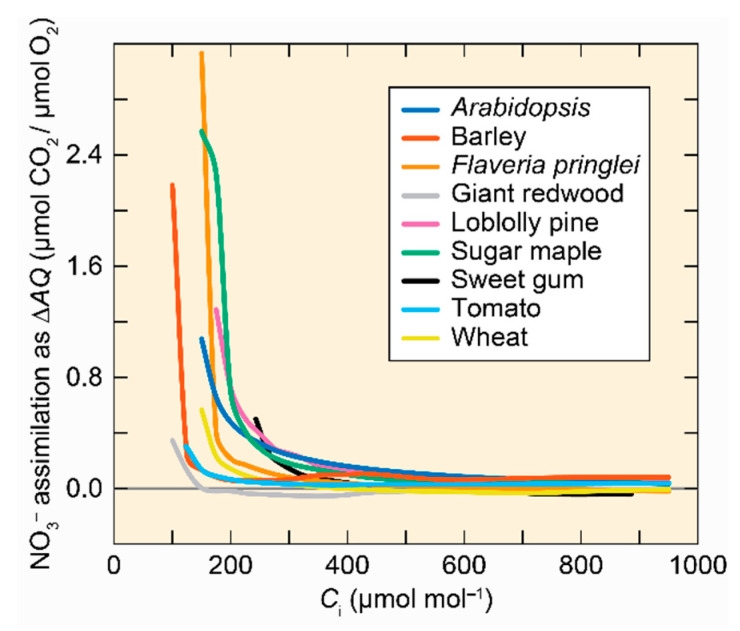
Shoot NO_3_^–^ assimilation as a function of shoot internal CO_2_ concentration (*C*_i_) for 9 C_3_ species. Adapted with permission from ref. [101]. Copyright 2014 Springer Science Business Media Dordrecht.

**Figure 6 plants-10-00908-f006:**
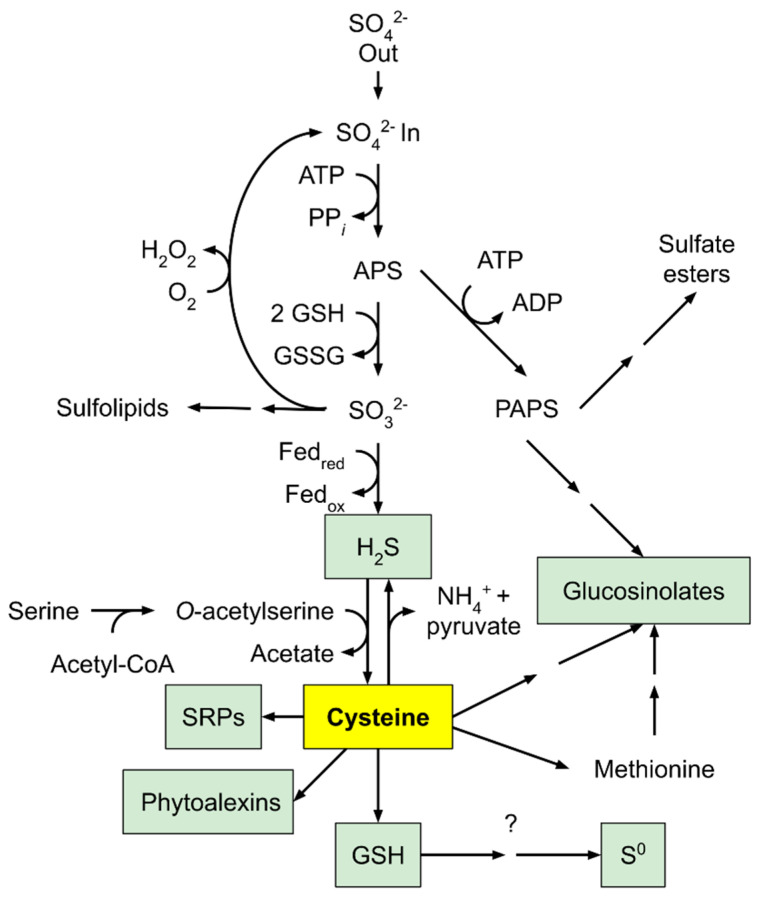
An outline of sulfur assimilation and its role in producing sulfur-containing defense compounds. Adapted with permission from ref. [137]. Copyright 2005 Elsevier Ltd.

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
