# Peer review of "Photorespiration: The Futile Cycle?"

_plants, 2021, doi:10.3390/plants10050908_

Round 1

Reviewer 1 Report

To the authors,
The authors summarized the significance and physiological importance of photorespiration in C3 plants on the basis of the author‘s great works. The reviewer always agrees with the statement that driving photorespiration is quite important for N assimilation, in addition to the electron sink function. The reviewer found that not only the summaries of physiological properties of photorespiration but also brief summaries of the method to evaluate photorespiration activities with some advantages and disadvantages. This manuscript would very valuable for publishing in “Plants”. As follows, the reviewer added some minor comments to this manuscript. Please check them.

Minor comments,
In several parts of the sentence, some subscriptions are not completed (Such as CO2 or NO3-). Please check the entire manuscript and please correct them.

Line 317 to 322; In this part, the authors described the method using GC-MS. However, in line 318, the reviewer understood that the following sentence carries the disadvantages of the LC-MS/MS method to quantify 2PG. Therefore, it might be difficult to make sense of the explanation of GC-MS in this part. The reviewer suggests that the disadvantages of the LC-MS/MS should be described before the explanation of GC-MS. 

Line 460; “Visualization, X.S” would be correct.

Figure 1; the description of “NO3- to protein” between “Rubisco” and “3PGA+2PG” might lead to misunderstanding to the reader. Please consider the rearrangement of this figure. 

Figure 2; More detailed description would be needed for readers.

Author Response

Dear Reviewer:

Thank you for taking the time to read this manuscript and provide critical comments and valuable suggestions.  Please see below for our response to your comments.

To the authors,
The authors summarized the significance and physiological importance of photorespiration in C3 plants on the basis of the author‘s great works. The reviewer always agrees with the statement that driving photorespiration is quite important for N assimilation, in addition to the electron sink function. The reviewer found that not only the summaries of physiological properties of photorespiration but also brief summaries of the method to evaluate photorespiration activities with some advantages and disadvantages. This manuscript would very valuable for publishing in “Plants”. As follows, the reviewer added some minor comments to this manuscript. Please check them.

Minor comments,
In several parts of the sentence, some subscriptions are not completed (Such as CO2 or NO3-). Please check the entire manuscript and please correct them.

We have checked and corrected the subscripts.

Line 317 to 322; In this part, the authors described the method using GC-MS. However, in line 318, the reviewer understood that the following sentence carries the disadvantages of the LC-MS/MS method to quantify 2PG. Therefore, it might be difficult to make sense of the explanation of GC-MS in this part. The reviewer suggests that the disadvantages of the LC-MS/MS should be described before the explanation of GC-MS. 

Thank you for pointing this out. We agree that our previous description of the mass spectrometry method was a bit confusing because it did not fully convey that both GC-MS and LC-MS/MS were used in an integrated mass spectrometry-based method. We have modified lines 317-341 slightly along with the caption of 3.2.7 for clarification.

Line 460; “Visualization, X.S” would be correct.

Line 460, now line 478, has been corrected.

Figure 1; the description of “NO3- to protein” between “Rubisco” and “3PGA+2PG” might lead to misunderstanding to the reader. Please consider the rearrangement of this figure. 

We have decided to remove the part of the figure labeled “NO3- to protein” between “Rubisco” and “3PGA+2PG” because it could cause confusion. The removal of this arrow should not affect the overall message figure 1 is trying to convey.

Figure 2; More detailed description would be needed for readers.

We have added section 2.3 at line 151-157 to better describe figure 2, and we have moved some of the content from section 2.2 into this section for better organization. We believe that a brief introduction on the interactions among different pathways displayed in figure 2 is sufficient for this review because we wanted to highlight the alternative (blue) pathway hypothesized.

Reviewer 2 Report

This seems to be a well-conducted analysis of the role of photorespiration in plants.  Photorespiration is like a black box, only a small group of researchers has the courage to study and bring new points of view, however, the topic of photorespiration is very attractive for plant research. 
The Paper is very competent. Photorespiratory metabolism allows plants to thrive in a high-oxygen-containing environment. Over the past years, the metabolic interaction of photorespiration and photosynthetic CO2 fixation has attracted major interest because research has demonstrated the enhancement of C3 photosynthesis and growth through the genetic manipulation of photorespiration.
Many authors (Cornic, Heber, Osmond, Brestic, Epron, Dreyer, Huang)  brought very interesting fundamental pieces of information about the role of photorespiration under different environmental stress.
The introduction provides a good understanding of the subject and its importance, with a significant quantity of information. Theoretical and practical reasons for the experiments are very reasonable. The authors attempted to assess the inter-organelle interaction during the photorespiratory pathway by induction of oxidative stress with the use of menadione in mitochondria under photo-oxidative stress.
I have no critical comments. Paper is very complex and innovative. Authors should describe better the importance of photorespiration under drought and high light, and also in C4 plants. Oxidative stress created in mitochondria causes a coordinative upregulation of photorespiration in other organelles at the transcriptional and translational levels and reactive oxygen species are important signals for inter-organelle communication during photorespiration.
The structure of the paper is logical and the results are well reproduced. The introduction and discussion are well organized. Results reported have not been published elsewhere. Conclusions are presented in an appropriate fashion and are supported by the data.
The analytical work of the authors is perfect, very well illustrated. I have no critical comments. Paper is very complex and innovative. Authors should include the missing information (research gaps and the significance of this kind of research) and add new pieces of information about the role of glutamine synthase in the regulation of photorespiration under stress conditions. 
The content of the manuscript is interesting and important for a wide range of readership in plant science research.
The discussion is sufficient and the conclusions are well formulated and justified.  I suggest adding also new references to support the interpretations. I appreciate the style of writing and illustrations of the paper. The authors are advised to read/cite the following papers related to their topics: https://doi.org/10.1016/j.envexpbot.2019.103845; http://dx.doi.org/10.1016/j.molp.2016.09.011; https://doi.org/10.1007/BF00203643; doi: 10.1111/j.1365-3040.2012.02567.x; 
The authors could add new more concrete conclusions, future perspectives. The results can promote future research.
I think the overall concept is interesting and potentially important. I recommend to ACCEPT the paper for publication with minor revision.

Author Response

Dear Reviewer:
Thank you for taking the time to read this manuscript and provide critical comments and valuable suggestions.  Please see below for our response to your comments.

This seems to be a well-conducted analysis of the role of photorespiration in plants.  Photorespiration is like a black box, only a small group of researchers has the courage to study and bring new points of view, however, the topic of photorespiration is very attractive for plant research. 
The Paper is very competent. Photorespiratory metabolism allows plants to thrive in a high-oxygen-containing environment. Over the past years, the metabolic interaction of photorespiration and photosynthetic CO2 fixation has attracted major interest because research has demonstrated the enhancement of C3 photosynthesis and growth through the genetic manipulation of photorespiration.
Many authors (Cornic, Heber, Osmond, Brestic, Epron, Dreyer, Huang)  brought very interesting fundamental pieces of information about the role of photorespiration under different environmental stress.
The introduction provides a good understanding of the subject and its importance, with a significant quantity of information. Theoretical and practical reasons for the experiments are very reasonable. The authors attempted to assess the inter-organelle interaction during the photorespiratory pathway by induction of oxidative stress with the use of menadione in mitochondria under photo-oxidative stress.
I have no critical comments. Paper is very complex and innovative. Authors should describe better the importance of photorespiration under drought and high light, and also in C4 plants. Oxidative stress created in mitochondria causes a coordinative upregulation of photorespiration in other organelles at the transcriptional and translational levels and reactive oxygen species are important signals for inter-organelle communication during photorespiration.

We acknowledge that a multitude of factors affect the balance of photosynthesis and photorespiration.  We have added a sentence (lines 467-474) regarding the importance of photorespiration as means of adaptation to various conditions such as drought and high light.  The connection between this adaptation and the biochemical processes that are the primary focus of our paper is one of the directions that we have suggested for future research.

The structure of the paper is logical and the results are well reproduced. The introduction and discussion are well organized. Results reported have not been published elsewhere. Conclusions are presented in an appropriate fashion and are supported by the data.
The analytical work of the authors is perfect, very well illustrated. I have no critical comments. Paper is very complex and innovative. Authors should include the missing information (research gaps and the significance of this kind of research) and add new pieces of information about the role of glutamine synthase in the regulation of photorespiration under stress conditions. 

We recognize the importance of glutamine synthetase in the regulation of photorespiration pathways under stress conditions. We believe such studies will definitely bring us closer to understanding the connections among nitrogen metabolism, photosynthesis and photorespiration pathways. It should be examined and discussed in detail in a separate review or a research paper.

The content of the manuscript is interesting and important for a wide range of readership in plant science research.
The discussion is sufficient and the conclusions are well formulated and justified.  I suggest adding also new references to support the interpretations. I appreciate the style of writing and illustrations of the paper. The authors are advised to read/cite the following papers related to their topics: https://doi.org/10.1016/j.envexpbot.2019.103845; http://dx.doi.org/10.1016/j.molp.2016.09.011; https://doi.org/10.1007/BF00203643; doi: 10.1111/j.1365-3040.2012.02567.x; 

We have read and added a few of the mentioned articles as references in the conclusion section. They are crucial findings to support the conclusion that the energy and chemicals produced by photorespiration are being used efficiently and not lost.

The authors could add new more concrete conclusions, future perspectives. The results can promote future research.

Several more lines of conclusions and future directions were added at line 467-475 based on the reviewers’ suggested references.

I think the overall concept is interesting and potentially important. I recommend to ACCEPT the paper for publication with minor revision.

Reviewer 3 Report

The manuscript is well written and can be accepted for publication.  Please elaborate Abstract and Conclusion sections.

Author Response

Dear Reviewer:
Thank you for taking the time to read this manuscript and provide critical comments and valuable suggestions.  Please see below for our response to your comments.

The manuscript is well written and can be accepted for publication.  Please elaborate Abstract and Conclusion sections.

We added more content in the conclusion section (line 467-475). Due to character limitations, we decided to keep the abstract section the same.